# Factor Graph-Based Online Bayesian Identification and Component Evaluation for Multivariate Autoregressive Exogenous Input Models [note 1]

**DOI:** 10.3390/e27070679

**Published:** 2025-06-26

**Authors:** Tim N. Nisslbeck, Wouter M. Kouw

**Affiliations:** Department of Electrical Engineering, Eindhoven University of Technology, 5612 AZ Eindhoven, The Netherlands; w.m.kouw@tue.nl

**Keywords:** Bayesian inference, probabilistic graphical models, message passing, system identification, stochastic systems, autoregressive models

## Abstract

We present a Forney-style factor graph representation for the class of multivariate autoregressive models with exogenous inputs, and we propose an online Bayesian parameter-identification procedure based on message passing within this graph. We derive message-update rules for (1) a custom factor node that represents the multivariate autoregressive likelihood function and (2) the matrix normal Wishart distribution over the parameters. The flow of messages reveals how parameter uncertainty propagates into predictive uncertainty over the system outputs and how individual factor nodes and edges contribute to the overall model evidence. We evaluate the message-passing-based procedure on (i) a simulated autoregressive system, demonstrating convergence, and (ii) on a benchmark task, demonstrating strong predictive performance.

## 1. Introduction

Autoregressive models provide a simple yet powerful framework for capturing dynamical systems [1,2,3,4,5]. Multivariate autoregressive models with exogenous inputs (MARX) exhibit a complex dependence structure. Each component of the vector signal evolves as a weighted combination of (i) its own past observations, (ii) other components, and (iii) an exogenous vector-valued input signal [6,7]. This intricate dependence structure generates significant uncertainty in parameter estimation.

Bayesian inference offers a principled approach for quantifying and propagating this uncertainty into predictions for future system outputs [8,9]. Moreover, uncertainty quantification enables the incorporation of information-theoretic quantities into cost functions, which is useful for optimal experimental design and adaptive control [10,11]. Markov Chain Monte Carlo techniques are typically employed to approximate posterior distributions. However their computational cost makes them impractical for large-scale real-time applications such as online system identification and adaptive control. In contrast, exact and variational inference methods provide full posterior distributions over parameters, thereby enabling robust decision-making under uncertainty [12,13]. This capability is particularly crucial in safety-critical applications, such as robotics, where understanding uncertainty is as important as making accurate predictions.

To address this challenge, we introduce an exact recursive Bayesian estimator that maintains a full posterior distribution and is computationally efficient. Recursive estimators offer a scalable alternative to batch estimators, but they either lack posterior uncertainty over parameters or rely on approximations [3,8]. Shaarawy and Ali proposed an exact recursive Bayesian estimator based on the matrix normal Wishart distribution, demonstrating its effectiveness for system identification [9]. We extend their approach by casting the inference procedure as a message-passing algorithm on a factor graph, thereby improving both computational efficiency and interpretability.

Factor graphs are graphical tools that capture the probabilistic relationships between random variables [14]. Many algorithms, including inference, can be formulated as message passing on a factor graph. Thus, message passing on factor graphs provides a structured and scalable framework for Bayesian inference, offering several key advantages over conventional inference frameworks [15,16,17]. We specifically consider Forney-style factor graphs, for their simplicity and compact visual representation [18]. First, factor graphs offer an intuitive representation of probabilistic models and data flow by depicting distinct probabilistic relationships as separate factor nodes that explicitly capture dependencies between variables [15,17]. This structured representation makes the inference process more interpretable and supports a more flexible model design, contributing to explainable artificial intelligence [19,20]. Second, message passing on factor graphs enables distributed computation by structuring inference into localized update rules at each node [21]. In particular, casting inference as message passing on a factor graph can enable federated learning, which accelerates learning in a multi-agent setting where physically separated agents share likelihood messages for joint parameter estimation [22]. This formulation significantly reduces the computational complexity compared to traditional recursive methods, making real-time Bayesian inference more tractable in large-scale settings [23,24]. Localized updates facilitate the efficient propagation of uncertainty throughout the graph, allowing for the attribution of uncertainty to specific sources, for example, distinguishing between prediction uncertainty arising from the likelihood model versus uncertainty in the inferred parameters. This fine-grained decomposition of uncertainties further enables a novel evaluation of model performance: the negative log-model evidence (surprisal) can be decomposed into contributions from individual nodes and edges in the factor graph. By analyzing how these contributions evolve over time, one gains detailed insights into the learning dynamics during system identification, thus linking model evaluation directly to the underlying probabilistic structure. Lastly, message passing unifies a broad class of algorithms, spanning signal filtering, optimal control, and path planning [14,17,24,25], making it a computationally efficient tool for probabilistic reasoning in large-scale problems. Overall, by leveraging this structured inference technique, our approach not only enhances Bayesian inference for dynamical systems but also yields more interpretable, scalable, and computationally efficient probabilistic machine learning models.

In summary, our key contributions are as follows:We derive a message-passing algorithm for exact recursive Bayesian inference in MARX models, maintaining full posterior distributions while ensuring computational efficiency.We extend the inference framework to predict future system outputs that explicitly account for parameter uncertainty, improving robustness for real-time applications.We introduce a novel model evaluation method by decomposing the negative log-model evidence (surprisal) into contributions from individual nodes and edges in the factor graph, providing insights into uncertainty and learning dynamics.We demonstrate the effectiveness of our approach through empirical evaluations on (i) a synthetic MARX system with known parameters for verification, and (ii) two synthetic dynamical systems with unknown parameters: a double mass-spring-damper system and a nonlinear double pendulum system.

The remainder of this paper is organized as follows. In Section 2, we formally describe the class of the discrete-time dynamical system considered. In Section 3, we present our probabilistic MARX model and its representation using Forney-style factor graphs. In Section 4, we detail the message-passing algorithm for recursive Bayesian inference, including both parameter estimation and predictive inference. In Section 5, we introduce our novel evaluation method based on decomposing surprisal. In Section 6, we demonstrate the effectiveness of our approach on synthetic system identification tasks. In Section 7, we discuss the computational benefits, interpretability, and broader implications of our method. Finally, in Section 8, we conclude this paper.

## 2. Problem Statement

We consider discrete-time dynamical systems, represented by a state $z_{k}\in R^{D_{z}}$ and driven by a control signal $u_{k}\in R^{D_{u}}$. These systems evolve according to a state transition function $f:R^{D_{z}}\times R^{D_{u}}\mapsto R^{D_{z}}$. At each time step, we observe a noisy measurement $y_{k}\in R^{D_{y}}$ of the state via a measurement function $g:R^{D_{z}}\mapsto R^{D_{y}}$. This can be expressed as a state–space model of the form: $$z_{k}=f\left(z_{k-1},u_{k}\right)\;,\;y_{k}=g\left(z_{k}\right)+e_{k}\;,$$
where $e_{k}\in R^{D_{y}}$ is a stochastic disturbance. Our objective is to predict future observations $y_{t}$ for t>k, given future inputs $u_{t}$, without prior knowledge about the system dynamics.

## 3. Model Specification

To address the problem defined in Section 2, we propose a probabilistic model that enables recursive learning and prediction of future observations in a partially observed dynamical system. Specifically, we assume that the unknown system can be approximated by a multivariate autoregressive model with exogenous inputs of order *N*, denoted as MARX(*N*). Let $y_{k}\in R^{D_{y}}$ denote the $D_{y}$-dimensional observation at time step *k*. We collect the past $N_{y}$ outputs into the matrix$$\begin{array}{cc} {\bar{y}}_{k-1} & ≜\left[\begin{array}{cccc} y_{k-1,1} & y_{k-2,1} & \ldots & y_{k-N_{y},1}\\ \vdots & \ldots & \ldots & \vdots \\ y_{k-1,D_{y}} & y_{k-2,D_{y}} & \ldots & y_{k-N_{y},D_{y}} \end{array}\right]\;, \end{array}$$
and, similarly, the most recent $N_{u}$ control inputs into$$\begin{array}{cc} {\bar{u}}_{k} & ≜\left[\begin{array}{cccc} u_{k,1} & u_{k-1,1} & \ldots & u_{k-N_{u}+1,1}\\ \vdots & \ldots & \ldots & \vdots \\ u_{k,D_{u}} & u_{k-1,D_{u}} & \ldots & u_{k-N_{u}+1,D_{u}} \end{array}\right]\;. \end{array}$$

We then reshape both matrices ${\bar{y}}_{k-1}$ and ${\bar{u}}_{k}$ into a single vector $x_{k}\in R^{D_{x}}$, where $D_{x}=N_{y}D_{y}+N_{u}D_{u}$:(1)$$\begin{array}{c} x_{k}≜\left[\begin{array}{c} \mathrm{vec}({\bar{y}}_{k-1})\\ \mathrm{vec}({\bar{u}}_{k}) \end{array}\right]\;, \end{array}$$
and vec(·) denotes the column-wise vectorization operator that stacks the columns of a matrix into a single column vector [26]. At the core of our MARX(*N*) model is a vector autoregressive process with exogenous inputs, characterized by the following likelihood function:(2)$$\begin{array}{cc} p(y_{k}\;|\;\Theta ,x_{k}) & =N(y_{k}\;|\;A^{⊺}x_{k},W^{-1})\\ \; & =\sqrt{\frac{\left|W\right|}{{\left(2\pi \right)}^{D_{y}}}}\exp\left(-\frac{1}{2}{\left(y_{k}-A^{⊺}x_{k}\right)}^{⊺}W\left(y_{k}-A^{⊺}x_{k}\right)\right)\;, \end{array}$$
where the parameters—jointly denoted as Θ=(A,W)—consist of a regression coefficient matrix $A\in R^{D_{x}\times D_{y}}$ and a noise precision matrix $W\in R_{+}^{\;D_{y}\times D_{y}}$, with $R_{+}$ denoting the space of positive semi-definite matrices. Each column $A_{:,j}$ specifies how the full memory vector $x_{k}$ (comprising past outputs and inputs) linearly predicts the *j*th component of the current observation $y_{k,j}$. In state–space terminology, *A* captures both the temporal memory and cross-variable coupling by weighting each lagged signal in $x_{k}$. The matrix *W* represents the inverse covariance (precision) of the Gaussian measurement noise: its diagonal entries set the inverse variances for each observed dimension while off-diagonals model instantaneous noise correlations between different components of $y_{k}$.

For computational convenience (see Section 4.1), we specify our prior distribution over Θ as a matrix normal Wishart distribution [27]:(3)$$\begin{array}{c} p\left(\Theta \right)=p\left(A\;|\;W\right)p\left(W\right)=\mathrm{MN}\left(A\;|\;M_{0},\Lambda_{0}^{-1},W^{-1}\right)W\left(W\;|\;\Omega_{0}^{-1},\nu_{0}\right)\;. \end{array}$$

Here, the coefficient matrix *A* follows a matrix normal distribution with mean $M_{0}\in R^{D_{x}\times D_{y}}$, row covariance $\Lambda_{0}^{-1}\in R^{D_{x}\times D_{x}}$, and column covariance $W^{-1}\in R^{D_{y}\times D_{y}}$,(4)$$\begin{array}{cc} p(A\;|\;W)\; & =\mathrm{MN}(A\;\;|\;M_{0},\Lambda_{0}^{-1},W^{-1})\\ \; & =\sqrt{\frac{{\left|W\right|}^{D_{x}}{\left|\Lambda_{0}\right|}^{D_{y}}}{{\left(2\pi \right)}^{D_{x}D_{y}}}}\exp\left(-\frac{1}{2}\mathrm{tr}\left[W{\left(A-M_{0}\right)}^{⊺}\Lambda_{0}\left(A-M_{0}\right)\right]\right)\;, \end{array}$$
where tr(·) denotes the trace of a square matrix, i.e., the sum of its diagonal entries [26]. The precision matrix *W* follows a Wishart distribution with a scale matrix $\Omega_{0}^{-1}\in R^{D_{y}\times D_{y}}$ and degrees of freedom $\nu_{0}\in R$
$$\begin{array}{c} p\left(W\right)=W\left(W\;|\;\Omega_{0}^{-1},\nu_{0}\right)=\sqrt{\frac{|\Omega_{0}{|}^{\nu_{0}}}{2^{\nu_{0}D_{y}}}}\frac{\sqrt{{\left|W\right|}^{\nu_{0}-D_{y}-1}}}{\Gamma_{D_{y}}\left(\nu_{0}/2\right)}\exp\left(-\frac{1}{2}\mathrm{tr}\left[W\Omega_{0}\right]\right)\;. \end{array}$$

Here, $\Gamma_{D_{y}}\left(\cdot \right)$ is the multivariate Gamma function with dimension $D_{y}$ [28]. Our goal is to infer the posterior distribution over *A* and *W* and subsequently use these parameter posterior distributions to make predictions for future outputs $y_{t}$.

The chosen prior and likelihood define the following generative model over the joint distribution of observations, inputs, and parameters:$$p\left(y_{1:k},u_{1:k},\Theta \right)=p\left(\Theta \right)\prod_{i=1}^{k}p\left(y_{i}\;|\;\Theta ,x_{i}\right)\;.$$

We consider two inference paradigms for parameter estimation [29]. In *batch estimation*, the full dataset is used to compute the posterior:$$p\left(\Theta \;|\;y_{1:k},u_{1:k}\right)\propto p\left(\Theta \right)\prod_{i=1}^{k}p\left(y_{i}\;|\;\Theta ,x_{i}\right)\;.$$

Alternatively, in *recursive estimation*, the posterior is updated incrementally as new data arrives:$$p\left(\Theta \;|\;y_{1:k},u_{1:k}\right)\propto p\left(\Theta \;|\;y_{1:k-1},u_{1:k-1}\right)p\left(y_{k}\;|\;\Theta ,y_{1:k-1},u_{1:k}\right)\;.$$

In this paper, we focus on the recursive formulation, which enables efficient online model updates and is well suited for real-time applications and systems where storing and reprocessing the entire history is infeasible.

### Factor Graph

The probabilistic graphical model underlying the recursive formulation is straightforward, consisting of a prior distribution and a likelihood function. Figure 1 presents a Forney-style factor graph in which nodes represent factors, edges denote variables, and each edge connects exactly two nodes [15]. In the graph, time flows from left to right, predictions flow from top to bottom, and corrections flow from bottom to top. The factor node labeled MNW represents the matrix normal Wishart prediction distribution along with its associated prior parameters. The dashed box represents the composite likelihood node, which comprises (i) the concatenation operation described in (Equation 1), (ii) the dot–product operation between the regression coefficient matrix *A* and the memory $x_{k}$, and (iii) the stochastic disturbance. The equality node connects the parameters Θ to the likelihood nodes for each time step *k*.

## 4. Inference

Inference consists of two stages: (i) parameter estimation, where we infer model parameters from observed outputs $y_{k}$ (Section 4.1), and (ii) output prediction, where we forecast future outputs $y_{t}$ for t>k, given future system inputs $u_{k+1}$ (Section 4.2).

### 4.1. Parameter Estimation

We wish to recursively estimate the posterior distribution over the model parameters:$$\begin{array}{c} p\left(\Theta \;|\;D_{k}\right)=\frac{p(y_{k}|\;\Theta ,x_{k})}{p(y_{k}|\;u_{k},D_{k-1})}\;p\left(\Theta \;|\;D_{k-1}\right)\;, \end{array}$$
where $D_{k}={\left\{y_{i},u_{i}\right\}}_{i=1}^{k}$ denotes the data up to time *k*. Note that the memory vector $x_{k}$ is a subset of $D_{k-1}$. The evidence term in the denominator is(5)$$p\left(y_{k}\;|\;u_{k},D_{k-1}\right)=\int p\left(y_{k}\;|\;\Theta ,x_{k}\right)\;p\left(\Theta \;|\;D_{k-1}\right)\;d\Theta \;.$$

This evidence term will be discussed in detail in Section 5.

**Lemma** **1.**
*Combining the MARX likelihood (Equation 2) with a matrix normal Wishart prior distribution over MARX coefficient matrix A and precision matrix W (Equation 3) yields a matrix normal Wishart distribution:*

$$p\left(\Theta \;|\;D_{k}\right)=\mathrm{MNW}\left(A,W\;|\;M_{k},\Lambda_{k}^{-1},\Omega_{k}^{-1},\nu_{k}\right),$$

*with the following parameter updates:*

$$\begin{array}{cc} \nu_{k} & =\nu_{k-1}+1\\ \Lambda_{k} & =\Lambda_{k-1}+x_{k}x_{k}^{⊺}\\ M_{k} & ={\left(\Lambda_{k-1}+x_{k}x_{k}^{⊺}\right)}^{-1}\left(\Lambda_{k-1}M_{k-1}+x_{k}y_{k}^{⊺}\right)\\ \Omega_{k} & =\;\Omega_{k-1}\;+\;y_{k}y_{k}^{⊺}\;+\;M_{k-1}^{⊺}\Lambda_{k-1}M_{k-1}\;\\ \; & \;-\;{\left(\Lambda_{k-1}M_{k-1}\;+\;x_{k}y_{k}^{⊺}\right)}^{⊺}{\left(\Lambda_{k-1}\;+\;x_{k}x_{k}^{⊺}\right)}^{-1}\left(\Lambda_{k-1}M_{k-1}\;+\;x_{k}y_{k}^{⊺}\right)\;. \end{array}$$



See Appendix A for the proof. This solution can be cast as a message-passing procedure on a factor graph, allowing distributed computation [15,30].

In Figure 1, circled messages indicate the information flow between the factor nodes along the edges. Message ① represents the previous posterior belief over Θ=(A,W): (6)$$\begin{array}{c} \vec{①}=p\left(\Theta \;|\;D_{k-1}\right)=\mathrm{MNW}\left(A,W\;|\;M_{k-1},\Lambda_{k-1}^{-1},\Omega_{k-1}^{-1},\nu_{k-1}\right)\;. \end{array}$$

The sum–product message from the composite MARX likelihood towards its parameters is the likelihood function itself, re-expressible as a probability distribution over Θ.

**Lemma** **2.**
*The message from the composite MARX likelihood (Equation 2) towards its parameters is matrix normal Wishart distributed as follows:*

(7)
$$\begin{array}{cc} ↑② & =p\left(y_{k}|\;\Theta ,x_{k}\right)\propto \mathrm{MNW}\left(A,W|\;{\bar{M}}_{k},{\bar{\Lambda }}_{k}^{-1},{\bar{\Omega }}_{k}^{-1},{\bar{\nu }}_{k}\right)\;. \end{array}$$

*Its parameters are*

$$\begin{array}{cc} {\bar{\nu }}_{k} & =2-D_{x}+D_{y}\;,\;{\bar{\Lambda }}_{k}=x_{k}x_{k}^{⊺}\;,\\ {\bar{M}}_{k} & ={\left(x_{k}x_{k}^{⊺}\right)}^{-1}x_{k}y_{k}^{⊺}\;,\;{\bar{\Omega }}_{k}=0_{D_{y}\times D_{y}}\;. \end{array}$$



See Appendix B for the proof. Note that the scale matrix is not positive-definite, which implies that message ② is an improper distribution. Utilizing improper distributions is not uncommon when messages are intermediate results. For example, in variational and particle-based message passing, the messages are unnormalized and therefore also technically improper distributions [31,32]. However, should one want to visualize message ② or convert it to a related distribution, for instance, then the scale matrix can be perturbed with a machine precision offset (i.e., ${\bar{\Omega }}_{k}=10^{-8}\cdot I_{D_{y}\times D_{y}}$).

Message ③ results from multiplying messages ① and ② at the equality node [15].

**Lemma** **3.***Let $p_{1}$ and $p_{2}$ be two matrix normal Wishart distributions over the same random variables* Θ*:*
$$\begin{array}{cc} p_{1}\left(\Theta \right) & =\mathrm{MNW}(A,W\;|\;M_{1},\Lambda_{1}^{-1},\Omega_{1}^{-1},\nu_{1})\\ p_{2}\left(\Theta \right) & =\mathrm{MNW}(A,W\;|\;M_{2},\Lambda_{2}^{-1},\Omega_{2}^{-1},\nu_{2})\;. \end{array}$$*Their product is proportional to another matrix normal Wishart distribution:*
$$p_{1}\left(\Theta \right)p_{2}\left(\Theta \right)\propto \mathrm{MNW}\left(A,W\;|\;M_{3},\Lambda_{3}^{-1},\Omega_{3}^{-1},\nu_{3}\right),$$*and its parameters are combinations of $p_{1},p_{2}$’s parameters,*
$$\begin{array}{cc} \nu_{3} & =\nu_{1}+\nu_{2}+D_{x}-D_{y}-1\;,\\ \Lambda_{3} & =\Lambda_{1}+\Lambda_{2}\;,\\ M_{3} & ={\left(\Lambda_{1}+\Lambda_{2}\right)}^{-1}\left(\Lambda_{1}M_{1}+\Lambda_{2}M_{2}\right)\;,\\ \Omega_{3} & =\Omega_{1}+\Omega_{2}+M_{1}^{⊺}\Lambda_{1}M_{1}+M_{2}^{⊺}\Lambda_{2}M_{2}\\ \; & \;-{\left(\Lambda_{1}M_{1}+\Lambda_{2}M_{2}\right)}^{⊺}{\left(\Lambda_{1}+\Lambda_{2}\right)}^{-1}\left(\Lambda_{1}M_{1}+\Lambda_{2}M_{2}\right). \end{array}$$

See Appendix C for the proof.

**Theorem** **1.**
*The outgoing message from the equality node is proportional to the exact recursive posterior distribution:*

$$\vec{③}=\vec{①}\cdot ②↑\propto \mathrm{MNW}\left(A,W\;|\;M_{k},\Lambda_{k}^{-1},\Omega_{k}^{-1},\nu_{k}\right).$$



**Proof.** Combining parameters from the messages in (Equation 6) and (Equation 7) according to the product operation in Lemma 3 yields$$\begin{array}{cc} \nu_{k} & =\nu_{k-1}+{\bar{\nu }}_{k}+D_{x}-D_{y}-1=\nu_{k-1}+1\;,\\ \Lambda_{k} & =\Lambda_{k-1}+{\bar{\Lambda }}_{k}=\Lambda_{k-1}+x_{k}x_{k}^{⊺}\;,\\ M_{k} & ={\left(\Lambda_{k-1}+{\bar{\Lambda }}_{k}\right)}^{-1}\left(\Lambda_{k-1}M_{k-1}+{\bar{\Lambda }}_{k}{\bar{M}}_{k}\right)={\left(\Lambda_{k-1}+x_{k}x_{k}^{⊺}\right)}^{-1}\left(\Lambda_{k-1}M_{k-1}+x_{k}y_{k}^{⊺}\right)\;,\\ \Omega_{k} & =\Omega_{k-1}+{\bar{\Omega }}_{k}+M_{k-1}^{⊺}\Lambda_{k-1}M_{k-1}+{\bar{M}}_{k}^{⊺}{\bar{\Lambda }}_{k}{\bar{M}}_{k}\\ \; & \;-{\left(\Lambda_{k-1}M_{k-1}+{\bar{\Lambda }}_{k}{\bar{M}}_{k}\right)}^{⊺}{\left(\Lambda_{k-1}+{\bar{\Lambda }}_{k}\right)}^{-1}\left(\Lambda_{k-1}M_{k-1}+{\bar{\Lambda }}_{k}{\bar{M}}_{k}\right)\\ \; & =\Omega_{k-1}+M_{k-1}^{⊺}\Lambda_{k-1}M_{k-1}+y_{k}y_{k}^{⊺}\\ \; & \;-{\left(\Lambda_{k-1}M_{k-1}+x_{k}y_{k}^{⊺}\right)}^{⊺}{\left(\Lambda_{k-1}+x_{k}x_{k}^{⊺}\right)}^{-1}\left(\Lambda_{k-1}M_{k-1}+x_{k}y_{k}^{⊺}\right). \end{array}$$These match the parameter update rules outlined in Lemma 1. □

### 4.2. Output Prediction

Predicting future system outputs amounts to computing the posterior predictive distribution, i.e., the marginal distribution of $y_{t}$ for t>k: (8)$$↓④=p\left(y_{t}\;|\;u_{t},D_{k}\right)=\int p\left(y_{t}\;|\;\Theta ,x_{t}\right)p\left(\Theta \;|\;D_{k}\right)\;d\Theta \;.$$

We exploit the factorization of the parameter posterior over (A,W) to split this into a marginalization over *A*:$$p\left(y_{t}\;|\;W,u_{t},D_{k}\right)=\int p\left(y_{t}\;|\;\Theta ,x_{t}\right)p\left(A\;|\;W,D_{k}\right)\;dA\;,$$
and a marginalization over *W*:$$p\left(y_{t}\;|\;u_{t},D_{k}\right)=\int p\left(y_{t}\;|\;W,u_{t},D_{k}\right)p\left(W\;|\;D_{k}\right)\;dW\;.$$

**Theorem** **2.**
*Marginalizing the composite MARX likelihood (Equation 2) over the matrix normal distribution (Equation 4) for A yields a multivariate normal distribution:*

$$\begin{array}{cc} \int N(y_{t} & |\;A^{⊺}x_{t},W^{-1})\mathrm{MN}\left(A\;|\;M_{k},\Lambda_{k}^{-1},W^{-1}\right)\;dA=N\left(y_{t}\;|\;M_{k}^{⊺}x_{t},{\left(\lambda_{t}W\right)}^{-1}\right)\;, \end{array}$$

*where $\lambda_{t}≜{\left(1+x_{t}^{⊺}\Lambda_{k}^{-1}x_{t}\right)}^{-1}$.*


See Appendix D for the proof.

**Theorem** **3.**
*Marginalizing a multivariate normal distribution over a Wishart distribution on its precision parameter yields a multivariate location-scale Student’s t-distribution [27]:*

(9)
$$\begin{array}{cc} \int N(y_{t} & |\;M_{k}^{⊺}x_{t},{\left(\lambda_{t}W\right)}^{-1})W\left(W|\;\Omega_{k}^{-1},\nu_{k}\right)\;dW=T\left(y_{t}|\;\mu_{t},\Psi_{t}^{-1},\eta_{t}\right)\;, \end{array}$$

*where $\mu_{t}≜M_{k}^{⊺}x_{t}$, $\eta_{t}≜\nu_{k}-D_{y}+1$, and $\Psi_{t}≜\eta_{t}\Omega_{k}^{-1}\lambda_{t}$.*


See Appendix E for the proof. The resulting posterior predictive distribution provides a recursive estimate of output uncertainty, which is valuable for decision-making and adaptive control.

## 5. Model Evaluation

A key criterion for probabilistic model evaluation is the negative log-model evidence (or surprisal) $-\log p(y_{k})$, which quantifies how surprising the observed data $y_{k}$ is under the model [33,34]. To gain deeper insights into model performance, we analyze surprisal from the perspective of variational inference on factor graphs. This approach enables us to decompose the overall model score into contributions from the individual nodes and edges of the graph.

Variational inference casts Bayesian inference as an optimization problem by approximating the true posterior $p(\Theta \;|\;D_{k})$ with a computationally tractable variational posterior $q(\Theta \;|\;D_{k})$, chosen from a variational family *Q* [33,35]. At time *k*, the optimal variational posterior is obtained by minimizing variational free energy (VFE) [36,37]:$$q^{*}\left(\Theta \;|\;D_{k}\right)=\arg\min_{q\in Q}\;F_{VFE}\left[\;q\left(\Theta \;|\;D_{k}\right),p\left(y_{k},\Theta \right)\;\right]\;,$$
where the VFE functional $F_{VFE}$ is defined as$$F_{VFE}\left[q\left(\Theta \;|\;D_{k}\right),p\left(y_{k},\Theta \right)\right]=\underset{\mathrm{Inference}\;\mathrm{Cost}}{\underset{︸}{D_{KL}\left[q\left(\Theta \;|\;D_{k}\right)||p\left(\Theta \;|\;D_{k}\right)\right]}}-\log\underset{\mathrm{Model}\;\mathrm{Evidence}}{\underset{︸}{p(y_{k}\;|\;u_{k},D_{k-1})}}\;.$$

In exact inference, where the true posterior is computed via Bayes’ rule, the inference cost becomes zero, and the VFE equals the exact surprisal. When exact inference is intractable, VFE is expressed in a different way. By absorbing the evidence term into the Kullback–Leibler (KL)-divergence, the product of the posterior and the evidence becomes the joint distribution of the generative model, which can be decomposed into a likelihood times prior distribution. This yields the decomposition of free energy into complexity and accuracy terms [37]:(10)$$\begin{array}{cc} D_{KL}[q(\Theta & |\;D_{k})||p\left(\Theta \;|\;D_{k}\right)]-\log p\left(y_{k}\;|\;u_{k},D_{k-1}\right)\\ \; & =E_{q(\Theta \;|\;D_{k})}\left[\log\frac{q(\Theta |\;D_{k})}{p(y_{k},\Theta |\;D_{k})}\right]\\ \; & =\underset{\mathrm{Complexity}}{\underset{︸}{D_{KL}\left[q\left(\Theta \;|\;D_{k}\right)||p\left(\Theta \;|\;D_{k-1}\right)\right]}}+\underset{\mathrm{Accuracy}}{\underset{︸}{H[q\left(\Theta \;|\;D_{k}\right),p\left(y_{k}\;|\;\Theta ,x_{k}\right)]}}\;, \end{array}$$
where complexity measures how much the variational posterior deviates from the prior, penalizing unnecessary deviations from prior knowledge and controlling overfitting. Accuracy quantifies the model’s ability to explain the observed data, expressed as the expected negative log-likelihood under the variational posterior. To refine this decomposition further, we introduce an auxiliary entropy term $H(\Theta \;|\;D_{k})$ and rewrite (Equation 10) as(11)$$\begin{array}{cc} \; & F_{VFE}\left[q\left(\Theta \;|\;D_{k}\right),p\left(y_{k},\Theta \right)\right]\\ \; & =\;D_{KL}\left[q\left(\Theta |D_{k}\right)∥p\left(\Theta |D_{k-1}\right)\right]\;+\;H\left[q\left(\Theta |D_{k}\right),p\left(y_{k}|\Theta ,x_{k}\right)\right)]\;-\;H\left[q\left(\Theta |D_{k}\right)\right]\;+\;H\left[q\left(\Theta |D_{k}\right)\right]\\ \; & =\;D_{KL}\left[q\left(\Theta \;|\;D_{k}\right)||p\left(\Theta \;|\;D_{k-1}\right)\right]+D_{KL}\left[q\left(\Theta \;|\;D_{k}\right)||p\left(y_{k}\;|\;\Theta ,x_{k}\right)\right)]+H\left[q\left(\Theta \;|\;D_{k}\right)\right]\;. \end{array}$$For models formulated as Forney-style factor graphs, inference is performed by optimizing the Bethe Free Energy (BFE), a generalization of VFE, which accounts for the graph’s structure [13,21,38]:(12)$$F_{BFE}\left[\;q\left(\Theta \;|\;D_{k}\right),p\left(y_{k},\Theta \right)\;\right]≜\sum_{a\in V}D_{KL}[q_{a}\left|\right|p_{a}]+\sum_{i\in E}H\left[q_{i}\right]\;,$$
where V is the set of factor nodes and E is the set of edges. In this formulation, each $q_{a}$ is the local variational belief at node *a*, $p_{a}$ is the corresponding exact local distribution, and each edge *i* contributes an entropy term $H[q_{i}]$. In our recursive MARX model—comprising a MARX likelihood node, a prior node, and an edge for the joint parameters Θ—the BFE decomposition in (Equation 12) coincides with the VFE decomposition in (Equation 11). Thus, factor graphs enable a fine-grained attribution of surprisal to specific components of the system.

### 5.1. MARX Model Evidence and Surprisal

To evaluate the model properly, we must compute the model evidence (marginal likelihood), which is the probability of an observed sample marginalized over parameters, weighted by their prior probabilities. Equation (Equation 5) already detailed the evidence term, but this still involved an integral. This integral is identical to the integral for the posterior predictive distribution (Equation 8), except that $y_{k}$ and $u_{k}$ are observed and the prior parameters are those from time step k−1. Concretely,$$\begin{array}{cc} \; & p\left(y_{k}\;|\;u_{k},D_{k-1}\right)=\int p\left(y_{k}\;|\;\Theta ,x_{k}\right)p\left(\Theta \;|\;D_{k-1}\right)\;d\Theta =T\left(y_{k}\;|\;m_{k},\Psi_{k}^{-1},\eta_{k}\right)\\ \; & =\sqrt{\frac{|\Psi_{k}|}{{\left(\eta_{k}\pi \right)}^{D_{y}}}}\frac{\Gamma_{D_{y}}\left(\left(\eta_{k}+D_{y}\right)/2\right)}{\Gamma_{D_{y}}\left(\left(\eta_{k}+D_{y}-1\right)/2\right)}{\left(1+\frac{1}{\eta_{k}}{\left(y_{k}-m_{k}\right)}^{⊺}\Psi_{k}\left(y_{k}-m_{k}\right)\right)}^{-(\eta_{k}+D_{y})/2}\;, \end{array}$$
where $m_{k}=M_{k-1}^{⊺}x_{k}$, $\eta_{k}=\nu_{k-1}-D_{y}+1$, $\Psi_{k}=\eta_{k}\Omega_{k-1}^{-1}\lambda_{k}$, and $\lambda_{k}={\left(1+x_{k}^{⊺}\Lambda_{k-1}^{-1}x_{k}\right)}^{-1}$. Here $T(\cdot \;|\;\mu ,\Sigma^{-1},\nu )$ denotes the multivariate Student’s *t*-distribution with location μ, scale $\Sigma^{-1}$, and degrees of freedom ν. Unlike the posterior predictive distribution, the model evidence is a scalar: higher values indicate that the model better explains the observed data. Hence, the surprisal for our model is(13)$$\begin{array}{cc} -\log p(y_{k}\;|\;u_{k},D_{k-1})\; & =-\frac{1}{2}\log\left|\Psi_{k}\right|+\frac{D_{y}}{2}\log\left(\eta_{k}\pi \right)\\ \; & \;-\log\Gamma_{D_{y}}\left(\frac{\eta_{k}+D_{y}}{2}\right)+\log\Gamma_{D_{y}}\left(\frac{\eta_{k}+D_{y}-1}{2}\right)\\ \; & \;+\frac{\eta_{k}+D_{y}}{2}\log\left(1+\frac{1}{\eta_{k}}{\left(y_{k}-m_{k}\right)}^{⊺}\Psi_{k}\left(y_{k}-m_{k}\right)\right)\;. \end{array}$$

### 5.2. MARX Variational Free Energy

**Lemma** **4.***Let q and p be two matrix normal Wishart distributions over the same random variables* Θ*, representing the posterior and prior, respectively:*
$$\begin{array}{cc} q(\Theta \;|\;D_{k}) & =\mathrm{MNW}(\Theta \;|\;M_{k},\Lambda_{k}^{-1},\Omega_{k}^{-1},\nu_{k})\\ p(\Theta \;|\;D_{k-1}) & =\mathrm{MNW}(\Theta \;|\;M_{k-1},\Lambda_{k-1}^{-1},\Omega_{k-1}^{-1},\nu_{k-1})\;. \end{array}$$*The differential cross-entropy $H[q\left(\Theta \;|\;D_{k}\right),p\left(\Theta \;|\;D_{k-1}\right)]$ of the posterior relative to the prior is*
$$\begin{array}{cc} H[q\left(\Theta |\;D_{k}\right),p\left(\Theta |\;D_{k-1}\right)]\; & =-\frac{1}{2}D_{y}\log|\Lambda_{k-1}|+\frac{1}{2}\left(\nu_{k-1}+D_{x}-D_{y}-1\right)\log\left|\Omega_{k}\right|\\ \; & \;-\frac{1}{2}\nu_{k-1}\log\left|\Omega_{k-1}\right|+\frac{1}{2}\left(D_{y}+1\right)D_{y}\log2+\frac{1}{2}D_{x}D_{y}\log\pi \\ \; & \;+\log\Gamma_{D_{y}}\left(\frac{\nu_{k-1}}{2}\right)-\frac{1}{2}\left(\nu_{k-1}+D_{x}-D_{y}-1\right)\psi_{D_{y}}\left(\frac{\nu_{k}}{2}\right)\\ \; & \;+\frac{1}{2}\nu_{k}\mathrm{tr}\left(\Omega_{k}^{-1}{\left(M_{k}-M_{k-1}\right)}^{⊺}\Lambda_{k-1}\left(M_{k}-M_{k-1}\right)\right)\\ \; & \;+\frac{1}{2}D_{y}\mathrm{tr}\left(\Lambda_{k}^{-1}\Lambda_{k-1}^{⊺}\right)+\nu_{k}\mathrm{tr}\left(\Omega_{k}^{-1}\Omega_{k-1}\right)\;. \end{array}$$

See Appendix F for the proof.

**Lemma** **5.**
*Consider the matrix normal Wishart posterior:*

$$q\left(\Theta \;|\;D_{k}\right)=\mathrm{MNW}\left(A,W\;|\;M_{k},\Lambda_{k}^{-1},\Omega_{k}^{-1},\nu_{k}\right)\;.$$


*Its (differential) entropy is*

(14)
$$\begin{array}{cc} H[q\left(\Theta |\;D_{k}\right)]\; & =-\frac{1}{2}D_{y}\log|\Lambda_{k}|+\frac{1}{2}\left(D_{x}-D_{y}-1\right)\log\left|\Omega_{k}\right|\\ \; & \;+\frac{1}{2}\left(D_{y}+1\right)D_{y}\log2+\frac{1}{2}D_{x}D_{y}\log\pi +\frac{1}{2}\left(D_{x}+\nu_{k}\right)D_{y}\\ \; & \;+\log\Gamma_{D_{y}}\left(\frac{\nu_{k}}{2}\right)-\frac{1}{2}\left(\nu_{k}+D_{x}-D_{y}-1\right)\psi_{D_{y}}\left(\frac{\nu_{k}}{2}\right)\;. \end{array}$$



See Appendix G for the proof.

**Lemma** **6.***Let q and p be two matrix normal Wishart distributions over the same random variables* Θ*, representing the posterior and prior, respectively:*
$$\begin{array}{cc} q(\Theta \;|\;D_{k}) & =\mathrm{MNW}(\Theta \;|\;M_{k},\Lambda_{k}^{-1},\Omega_{k}^{-1},\nu_{k})\\ p(\Theta \;|\;D_{k-1}) & =\mathrm{MNW}(\Theta \;|\;M_{k-1},\Lambda_{k-1}^{-1},\Omega_{k-1}^{-1},\nu_{k-1})\;. \end{array}$$*The KL-divergence $D_{KL}\left[q\left(\Theta \;|\;D_{k}\right)||p\left(\Theta \;|\;D_{k-1}\right)\right]$ of the posterior from the prior (complexity) is*
$$\begin{array}{cc} D_{KL}\left[q\left(\Theta \;|\;D_{k}\right)||p\left(\Theta \;|\;D_{k-1}\right)\right]\; & =\frac{1}{2}D_{y}\log\frac{|\Lambda_{k}|}{|\Lambda_{k-1}|}+\frac{1}{2}\nu_{k-1}\log\frac{|\Omega_{k}|}{|\Omega_{k-1}|}-\frac{1}{2}\left(D_{x}+\nu_{k}\right)D_{y}\\ \; & \;-\log\Gamma_{D_{y}}\left(\frac{\nu_{k}}{2}\right)+\log\Gamma_{D_{y}}\left(\frac{\nu_{k-1}}{2}\right)+\frac{1}{2}\left(\nu_{k}-\nu_{k-1}\right)\psi_{D_{y}}\left(\frac{\nu_{k}}{2}\right)\\ \; & \;+\frac{1}{2}\nu_{k}\mathrm{tr}\left(\Omega_{k}^{-1}{\left(M_{k}-M_{k-1}\right)}^{⊺}\Lambda_{k-1}\left(M_{k}-M_{k-1}\right)\right)\\ \; & \;+\frac{1}{2}D_{y}\mathrm{tr}\left(\Lambda_{k}^{-1}\Lambda_{k-1}^{⊺}\right)+\nu_{k}\mathrm{tr}\left(\Omega_{k}^{-1}\Omega_{k-1}\right)\;. \end{array}$$

See Appendix H for the proof.

**Lemma** **7.**
*Consider a matrix normal Wishart distribution q and a multivariate normal distribution p, representing the posterior and MARX likelihood:*

$$\begin{array}{cc} q(\Theta \;|\;D_{k}) & =\mathrm{MNW}(A,W\;|\;M_{k},\Lambda_{k}^{-1},\Omega_{k}^{-1},\nu_{k})\\ p(y_{k}\;|\;\Theta ,x_{k}) & =N(y_{k}\;|\;A^{⊺}x_{k},W^{-1})\;. \end{array}$$

*The differential cross-entropy $H[q\left(\Theta \;|\;D_{k}\right),p\left(y_{k}\;|\;\Theta ,x_{k}\right)]$ of the posterior relative to the likelihood (accuracy) is*

$$\begin{array}{cc} H[q\left(\Theta |\;D_{k}\right),p\left(y_{k}|\;\Theta ,x_{k}\right)]\; & =-\frac{1}{2}\psi_{D_{y}}\left(\frac{\nu_{k}}{2}\right)+\frac{1}{2}\log\left|\Omega_{k}\right|+\frac{1}{2}D_{y}\log\pi \\ \; & \;+\frac{1}{2}\nu_{k}{\left(y_{k}-M_{k}^{⊺}x_{k}\right)}^{⊺}\Omega_{k}^{-1}\left(y_{k}-M_{k}^{⊺}x_{k}\right)+\frac{1}{2}x_{k}^{⊺}\Lambda_{k}^{-1}x_{k}D_{y}\;. \end{array}$$



See Appendix I for the proof.

## 6. Experiments

We conducted three experiments: one verification experiment and two validation experiments (Code: https://github.com/biaslab/MDPI2025-MARX, accessed on 8 March 2025). In the verification experiment (Section 6.2), we tested whether the MARX estimator could identify a dynamical system with known parameters. In the validation experiments (Section 6.3), we assess the estimator’s performance on two complex dynamical systems with unknown parameters: a linear double mass-spring-damper system and a nonlinear double pendulum. In all the experiments, we compare the performance of the MARX estimator to a baseline approach.

### 6.1. Baseline Estimator

We compare against a recursive least squares (RLS) estimator [3]. Let ${\hat{A}}_{k}$ be a point estimate of the coefficient matrix based on the previous *k* data points, and let $P_{0}=I_{D_{x}}$ be an initial inverse sample covariance matrix. These matrices are updated at each time step according to$$\begin{array}{cc} P_{k} & =P_{k-1}-P_{k-1}x_{k}{\left(1+x_{k}^{⊺}P_{k-1}x_{k}\right)}^{-1}x_{k}^{⊺}P_{k-1}\\ {\hat{A}}_{k} & ={\hat{A}}_{k-1}+P_{k-1}x_{k}{\left(1+x_{k}^{⊺}P_{k-1}x_{k}\right)}^{-1}{\left(y_{k}-{\hat{A}}_{k-1}^{⊺}x_{k}\right)}^{⊺}\;. \end{array}$$
Note that this formulation corresponds to a forgetting factor of 1.0, meaning that older data points are not down-weighted. The system outputs are predicted with $y_{t}={\hat{A}}_{k}^{⊺}x_{t}$.

### 6.2. Verification

We perform a verification experiment on a MARX system with state $z_{k}=x_{k}$ (Equation 1), memory sizes $N_{y}=2,N_{u}=3$, and dimensions $D_{y}=D_{u}=2$. The system has true parameters $\tilde{\Theta }=\left(\tilde{A},\tilde{W}\right)$. It evolves according to $g\left(f\left(x_{k}\right)\right)={\tilde{A}}^{⊺}x_{k}$, where $\tilde{A}$ is the known coefficient matrix (see Figure 2). For each output dimension *i*, the lag-dependent coefficients were generated using a Butterworth low-pass filter (cutoff frequency 20 Hz) applied to that same dimension, while cross-dimensional coefficients were sampled from $N(0,0.1^{2})$ [39]. We chose the Butterworth filter because its maximally flat response in the passband ensures that signals below the cutoff frequency are transmitted with little distortion while attenuating higher-frequency components [40]. This makes it suitable for generating stable linear dynamics and mimicking the low-pass behavior often observed in physical dynamical systems—such as mechanical or electrical processes [41,42]—and is common in applications like audio and biomedical signal processing [41,43]. The disturbance follows $e_{k}∼N\left(0,{\tilde{W}}^{-1}\right)$ with precision matrix $\tilde{W}=\left[\begin{array}{cc} 300 & 100\\ 100 & 200 \end{array}\right]$.

We evaluated each estimator for training sizes $T_{\mathrm{train}}\in \left\{2^{l}|l\in \left\{2,3,4,5,6\right\}\right\}$, using Monte Carlo experiments with $N_{MC}=100$ runs. To learn the parameters, each estimator uses $T_{\mathrm{train}}$ state transitions, starting from state $z_{0}=0_{D_{z}}$. After training, each estimator is tested for $T_{\mathrm{test}}=100$ time steps, again starting from $z_{0}$ but with different control signals. For the MARX estimator, we compare two priors (see Table 1): uninformative (MARX-UI) and weakly informative (MARX-WI). The uninformative prior uses small precision values for $\Lambda_{0}$ and $\Omega_{0}$, corresponding to large prior variancesthat reflect minimal prior belief about the parameters. The weakly informative prior assigns higher precision (lower variance), introducing a mild preference for more stable parameter values while still letting the data dominate. In both cases, the degrees of freedom $\nu_{0}$ are kept minimal at $D_{y}+3$, just above the threshold for the Wishart distribution to be well defined, further reinforcing the limited informativeness of the prior. The weakly informative prior also encodes approximate prior knowledge about the observation noise. Specifically, the Wishart component p(W) has a mode at $\nu_{0}\Omega_{0}^{-1}=\left[\begin{array}{cc} 500 & 0\\ 0 & 500 \end{array}\right]$, which is of similar magnitude to the true noise precision $\tilde{W}$. In contrast, the uninformative prior sets $\Omega_{0}$ to much larger, placing its mode far from the true noise characteristics. Thus, the weakly informative prior softly incorporates domain knowledge about expected noise levels, improving convergence and stability in the early stages of recursive estimation. For each training size, we calculate the root mean squared error (RMSE),$$\begin{array}{c} \mathrm{RMSE}=\sqrt{\frac{1}{T_{\mathrm{test}}}\sum_{k=1}^{T_{\mathrm{test}}}{\left({\hat{y}}_{k}-y_{k}\right)}^{2}}\;, \end{array}$$
between the predicted output ${\hat{y}}_{k}$, i.e., the mean of the posterior predictive $p(y_{k}\;|\;u_{k},D_{k-1})$, and the true output $y_{k}$ for all $k\in T_{\mathrm{test}}$ evaluation steps.

Figure 3 shows the simulation errors for MARX-UI, MARX-WI, and RLS as a function of the training size. For small sample sizes, MARX-WI consistently outperforms RLS, while MARX-UI performs slightly worse. All three estimators converge to the same performance level as the training size increases.

Figure 4 focuses on a single Monte Carlo experiment with $T_{\mathrm{train}}=2^{6}$. It plots $\log(||\tilde{A}-A|{|}_{F})$, the log of the Frobenius norm between the true coefficient matrix $\tilde{A}$ and each estimate *A*. MARX-WI consistently yields better estimates of $\tilde{A}$ than MARX-UI and RLS. Although MARX-UI struggles during the first 25 time steps, it eventually produces a more accurate estimate of $\tilde{A}$ compared to RLS.

Unlike RLS, the MARX estimator also estimates the noise precision matrix *W*. Figure 5 shows $\log(||\tilde{W}-W|{|}_{F})$ for both MARX-WI and MARX-UI. MARX-WI consistently achieves more accurate estimates of $\tilde{W}$ than MARX-UI.

Figure 6 plots the negative log posterior probability of the true parameters $\tilde{\Theta }$ (lower is better), showing that the posterior concentrates sharply on the true values. As a probabilistic estimator, MARX also quantifies uncertainty in its estimates of $\tilde{A}$ and $\tilde{W}$ via the posterior precision (or scale) parameters. Figure 7 illustrates the evolution of MARX-WI’s estimates of *W* for a single run with $T_{\mathrm{train}}=2^{6}$. The ribbon represents one standard deviation around the mean. Initially, MARX-WI exhibits high uncertainty (large variance), which generally decreases over time. Because $\tilde{W}$ and *W* are symmetric, only the upper-triangular elements are shown.

Figure 8 (top) shows a heatmap of the difference $A-\tilde{A}$. To save space, we plot only a subset of the elements of *A*, marked by “X”. This subset includes the elements with the largest estimation errors and two randomly selected elements. Figure 8 (bottom) shows the evolution of these selected elements for the same Monte Carlo experiment run, with ribbons indicating one standard deviation around each mean estimate.

Furthermore, we apply the model score decomposition from Section 5 to evaluate our recursive MARX model. By tracking how surprisal and its constituent terms evolve, we obtain fine-grained insights into the model’s learning dynamics and uncertainty reduction. We can recall from (Equation 10) that surprisal decomposes into an accuracy term—given by the cross-entropy of the variational posterior relative to the likelihood, reflecting data fit—and a complexity term—given by the KL-divergence of the variational posterior from the prior, quantifying deviation from prior beliefs. Figure 9 illustrates this decomposition. In the early stages of model training, the complexity term (green) dominates overall surprisal (dashed blue), indicating substantial updates from the prior as the model learns the system parameters. As training progresses and the posterior stabilizes, the complexity term diminishes, and the accuracy term (red) becomes the main source of uncertainty. Spikes in overall surprisal during later stages align with spikes in the accuracy term, which we interpret as indicators of measurement outliers that temporarily degrade model fit.

Figure 10 complements this analysis by plotting the entropy of the variational posterior $q(\Theta \;|\;D_{k})$ over time. This highlights how quickly the inference procedure narrows the parameter space, providing insight into convergence speed and residual uncertainty in the model parameters.

We also demonstrate model evaluation using model evidence. Figure 11 shows the evolution of surprisal (lower is better) over time for MARX-WI and MARX-UI. This plot highlights that the prior choice matters only initially; with sufficient data, MARX-WI and MARX-UI converge to the same performance.

### 6.3. Validation

To evaluate the proposed method, we perform validation experiments on two distinct mechanical systems: a *linear* double mass-spring-damper system and a *nonlinear* double pendulum system. These testbeds span a range of dynamical complexity and are standard benchmarks for modeling and control tasks. Despite their differences, both systems share a common formulation as second-order dynamical systems expressed in first-order ODE form:$$I_{k}{\ddot{z}}_{k}=F\left(z_{k},{\dot{z}}_{k},u_{k}\right)\;,$$
where $z_{k}$ denotes generalized coordinates, ${\dot{z}}_{k}$ and ${\ddot{z}}_{k}$ are the first and second time derivatives of $z_{k}$, $u_{k}$ are the control inputs, $I_{k}$ is a (state-dependent) generalized inertia matrix, and *F* encodes the system-specific generalized forces (including passive dynamics and external control inputs). Time evolution is performed using a forward Euler integrator with a system-specific time step Δt:$$z_{k+1}=z_{k}+\Delta t{\dot{z}}_{k}\;\mathrm{and}\;{\dot{z}}_{k+1}={\dot{z}}_{k}+\Delta t{\ddot{z}}_{k}\;.$$

For both validation systems, we choose a disturbance $e_{k}∼N\left(0,{\tilde{W}}^{-1}\right)$ with a precision matrix $\tilde{W}=\left[\begin{array}{cc} 2000 & 1000\\ 1000 & 2000 \end{array}\right]$. The validation experiments follow the same procedure as the verification experiment: we perform Monte Carlo experiments with $N_{MC}=100$ runs with Δt=0.05, in which each estimator has $T_{\mathrm{train}}\in \left\{2^{l}|l\in \left\{2,3,4,5,6\right\}\right\}$ state transitions to learn the parameters (starting from state $z_{0}=0_{D_{z}}$), and we test each estimator with $T_{\mathrm{test}}=100$ transitions. However, we increase the memory sizes of the MARX model to $N_{y}=N_{u}=5$.

In the following, we describe each validation system individually, and then present the combined validation results.

#### 6.3.1. Linear System: Double Mass-Spring-Damper

The linear system consists of two masses: $m_{1}=1.0$ kg, connected to a fixed base by a spring and damper with stiffness $k_{1}=0.99$ and damping $c_{1}=0.4$, and $m_{2}=2.0$ kg, connected to $m_{1}$ via a second spring and damper with $k_{2}=0.8$ and $c_{2}=0.4$. The generalized coordinates $z_{k}\in R^{2}$ represent the displacements of each mass from the equilibrium, and the generalized inertia matrix is a constant: $I_{k}=\mathrm{diag}\left(m_{1},m_{2}\right)$, where diag(·) denotes a diagonal matrix with the given entries [26]. The generalized force function *F* combines the internal spring and damping forces with external inputs:$$F\left(z_{k},{\dot{z}}_{k},u_{k}\right)=Kz_{k}+C{\dot{z}}_{k}+u_{k}\;,$$
with the stiffness and damping matrices:$$\begin{array}{cc} K & =\left[\begin{array}{cc} -(k_{1}+k_{2}) & k_{2}\\ k_{2} & -k_{2} \end{array}\right],\;C=\left[\begin{array}{cc} -(c_{1}+c_{2}) & c_{2}\\ c_{2} & -c_{2} \end{array}\right]\;. \end{array}$$

#### 6.3.2. Nonlinear System: Double Pendulum

The nonlinear system is a planar double pendulum (also called an acrobot) with two links of lengths $l_{1}=1.0$ m and $l_{2}=1.0$ m and masses $m_{1}=1.0$ kg and $m_{2}=1.0$ kg, respectively. The generalized coordinates $z_{k}\in R^{2}$ represent the joint angles, and the generalized inertia matrix is captured implicitly through a structured nonlinear force formulation. The dynamics are governed by gravity and nonlinear velocity coupling, yielding$$\begin{array}{c} F\left(z_{k},{\dot{z}}_{k},u_{k}\right)=\mathrm{diag}\left(g\left(\frac{1}{2}m_{1}+m_{2}\right)l_{1},\;-\frac{1}{2}gm_{2}l_{2}\right)\sin\left(z_{k}\right)+J_{x}V{\dot{z}}_{k}^{2}+u_{k}\;, \end{array}$$
where *g* is gravitational acceleration, $J_{x}≜\frac{1}{2}m_{2}l_{1}l_{2}$, and *V* is the nonlinear velocity-coupling matrix:$$\begin{array}{c} V=\left[\begin{array}{cc} 0 & -\sin(z_{k,1}-z_{k,2})\\ \sin(z_{k,1}-z_{k,2}) & 0 \end{array}\right]\;. \end{array}$$

#### 6.3.3. Results

As in the verification experiment, Figure 12 shows the simulation errors for MARX-UI, MARX-WI, and RLS for both the double mass-spring-damper system Figure 12a) and the double pendulum system (Figure 12b). Convergence to stable performance is slower in both systems compared to the verification case. Nevertheless, both MARX variants outperform RLS and converge to similar levels of predictive performance. This confirms that the MARX model generalizes to more complex dynamical systems. As expected, the overall RMSE is higher for the nonlinear double pendulum system. A peak of performance loss is present for MARX-UI, which is more pronounced in the double mass-spring-damper system.

Figure 13 shows $\log(||\tilde{W}-W|{|}_{F})$ for both MARX-WI and MARX-UI for the validation systems. Initially, MARX-WI achieves better accuracy and lower variability than MARX-UI. Unlike in the verification setting, MARX-UI improves significantly over time and ultimately approaches similar estimation quality.

Figure 14 illustrates estimates of $\tilde{W}$ by MARX-WI for a single Monte Carlo experiment ($T_{\mathrm{train}}=2^{6}$) for both systems. The model struggles with learning and initially shows high uncertainty, followed by a sharp reduction as learning progresses. This reflects the challenge of inferring observation noise structure in nonlinear systems from limited data.

Figure 15 displays the evolution of MARX-WI’s surprisal and its decomposition into accuracy and complexity. The early learning phases show that surprisal reduction is dominated by decreasing model complexity. This trend is more difficult to sustain in the nonlinear system, where complexity remains elevated for longer. Later in training, fluctuations in surprisal are primarily driven by changes in accuracy.

Finally, Figure 16 shows the entropy of the variational posterior $q(\Theta \;|\;D_{k})$ for each validation system. In both systems, MARX-WI rapidly reduces entropy, indicating fast convergence to informative parameter regions despite the different complexities of the systems.

## 7. Discussion

The modular nature of the factor graph methodology provides substantial practical advantages. As demonstrated by Loeliger et al. [15], factor graphs facilitate the visual construction of complex algorithms by incorporating, eliminating, or merging established computational units. For example, the MARX model’s factor graph (Figure 1) could be extended to support time-varying parameters by introducing state transition factor nodes between the equality nodes over the parameters [24]. In multi-agent robotics, where sensors and actuators are spread across various platforms, each agent can update its local beliefs through message passing and share only the most informative summaries [44]. This targeted communication reduces bandwidth demands while enabling swift convergence to an accurate global model. Recent research highlights the importance of transmitting informative variational beliefs in multi-agent environments [22,45], facilitating scalable cooperative learning among heterogeneous agents. The resulting computational decentralization opens promising opportunities for federated system identification and coordination in multi-robot systems, especially when subject to privacy or bandwidth constraints [46,47,48].

### 7.1. Computational Efficiency

The dominant computational cost in our inference algorithm arises from the matrix inversion of Λ (Equation 4), which scales as $O(D_{x}^{3})$ in the worst case. We benchmarked the update rule computations on a Julia-based implementation running on an Apple Macbook M1, averaging over 1,000,000 runs. For a state dimension of $D_{x}=10$, updating the parameters for a single time step took approximately 2 nanoseconds (excluding garbage collection). Further computational savings are possible by adopting an information filter parameterization, where $\Xi_{k}$ (Equation 17) is stored instead of $M_{k}$ (Equation 3) [49]. This approach defers the matrix inversion until $M_{k}$ is explicitly needed, offering an efficiency boost, particularly in high-dimensional or resource-constrained scenarios.

### 7.2. Limitations

Despite its efficiency and modularity, our method has several limitations. First, it does not support fully Bayesian *k*-step ahead predictions. Computing joint posterior predictives over a longer horizon is intractable under the current formulation and is challenging as it requires marginalization over a (deeply) nested set of autoregressive coefficients. Second, the model is built on a linear multivariate autoregressive likelihood, which—while computationally efficient—limits its expressiveness. In systems characterized by strong nonlinearities, this assumption can lead to underfitting and reduced predictive performance. Lastly, although we explored both uninformative and weakly informative priors, the model remains sensitive to prior settings, particularly in data-scarce settings or during the early stages of recursive estimation. In these scenarios, poor prior choices can significantly degrade both convergence speed and final performance.

### 7.3. Future Work

Future work may explore extending the MARX framework to accommodate time-varying parameters by inserting state-transition factors between the equality nodes—analogous to prior work on univariate autoregressive models [24]. Another extension is to utilize the posterior distributions over the parameters to formulate a mutual information-based cost function for input signal design [10].

## 8. Conclusions

We presented a recursive Bayesian estimation procedure for multivariate autoregressive models with exogenous inputs. The method produces matrix-variate posterior distributions over both the model coefficients and the noise precision, allowing uncertainty to be explicitly propagated into future output predictions. We also demonstrated how these uncertainty estimates enable the analysis of individual factor nodes and edges within the model, making it possible to assess their contributions to the overall model score and to identify potential outliers. The ability to track sources of uncertainty online and evaluate their impact on output predictions is especially valuable for applications such as Bayesian optimal experimental design or information-theoretic adaptive control.

## Figures and Tables

**Figure 1 entropy-27-00679-f001:**
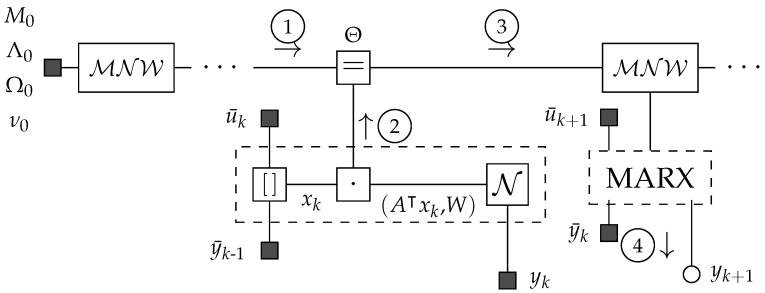
Forney-style factor graph of the MARX model in recursive form. A matrix normal Wishart node sends a prior message (1) to an equality node. A likelihood-based message (2) passes upwards from the MARX likelihood node (dashed box), attached to the observed variables $y_{k}$, ${\bar{y}}_{k-1}$, and ${\bar{u}}_{k}$. Combining the prior-based and likelihood-based messages at the equality node yields the posterior (message 3). Message 4 is the posterior predictive distribution for the system output.

**Figure 2 entropy-27-00679-f002:**
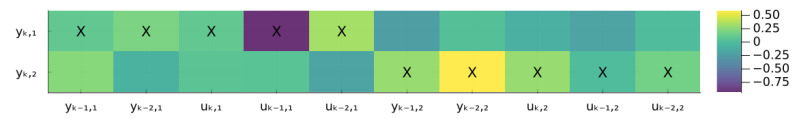
Heatmap of true system parameter ${\tilde{A}}^{⊺}$. “X” denotes coefficients generated from a Butterworth filter.

**Figure 3 entropy-27-00679-f003:**
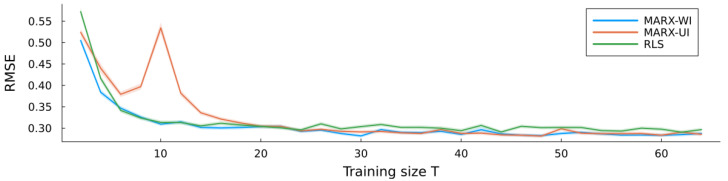
Simulation errors (average RMSE) of all three estimators for the MARX system, with ribbons indicating standard errors.

**Figure 4 entropy-27-00679-f004:**
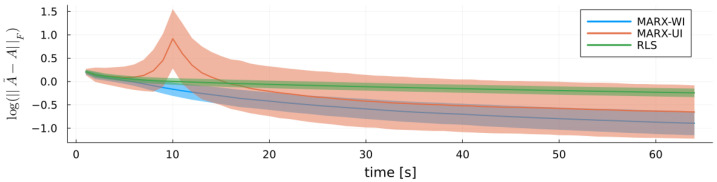
Log-scale Frobenius norm of the difference between true coefficient matrix $\tilde{A}$ and estimates *A* of each estimator in a single Monte Carlo run with $T_{\mathrm{train}}=2^{6}$ for the MARX system, with ribbons indicating standard errors.

**Figure 5 entropy-27-00679-f005:**
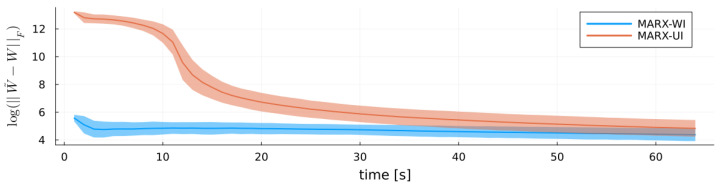
Log-scale Frobenius norm of the difference between true coefficient matrix $\tilde{W}$ and estimates *W* of each MARX estimator for the MARX system, with ribbons indicating standard errors.

**Figure 6 entropy-27-00679-f006:**
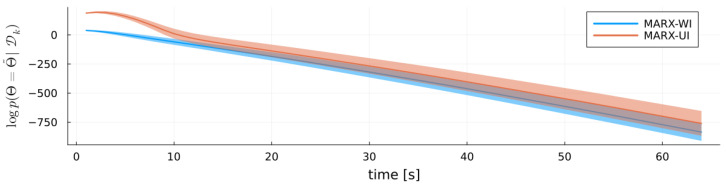
Negative log posterior probability of the true system parameters $\tilde{\Theta }$ under each prior choice for the MARX system (lower is better).

**Figure 7 entropy-27-00679-f007:**
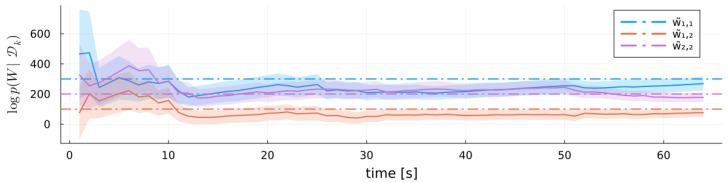
Time series of the estimated noise precision matrix *W* for the MARX-WI for the MARX system. Ribbons indicate one standard deviation, and horizontal lines denote the true values of $\tilde{W}$.

**Figure 8 entropy-27-00679-f008:**
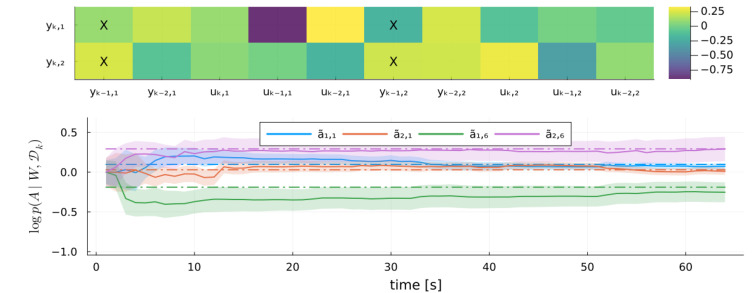
**Top**: Heatmap of the final ${\tilde{A}}^{⊺}$ coefficient matrix parameter estimate by the MARX-WI model. “X” marks selected elements, and the trajectories are shown below. **Bottom**: Time series of the selected elements of $\tilde{A}$ estimated by MARX-WI, with ribbons indicating one standard deviation. Horizontal lines show the true values of the corresponding elements of $\tilde{A}$.

**Figure 9 entropy-27-00679-f009:**
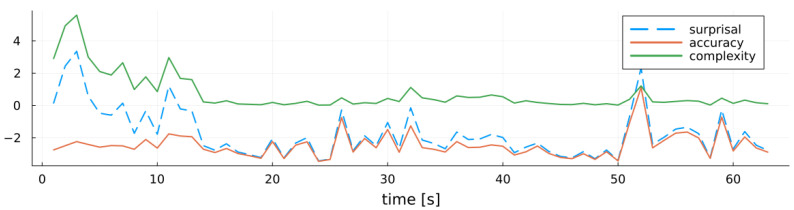
MARX-WI surprisal (dashed blue line) and its decomposition into accuracy (red line) and complexity (green line) over time for the MARX system.

**Figure 10 entropy-27-00679-f010:**
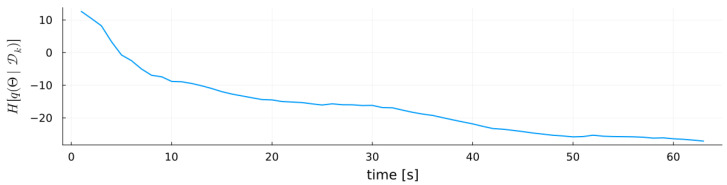
Entropy of the MARX-WI variational posterior $q(\Theta \;|\;D_{k})$ over time for the MARX system.

**Figure 11 entropy-27-00679-f011:**
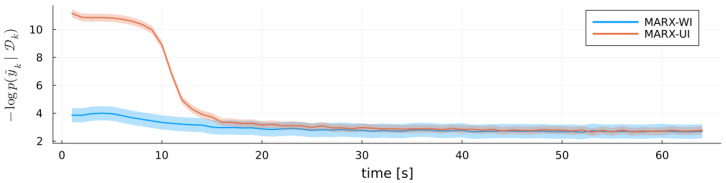
Surprisal over time for MARX-WI versus MARX-UI for the MARX system.

**Figure 12 entropy-27-00679-f012:**
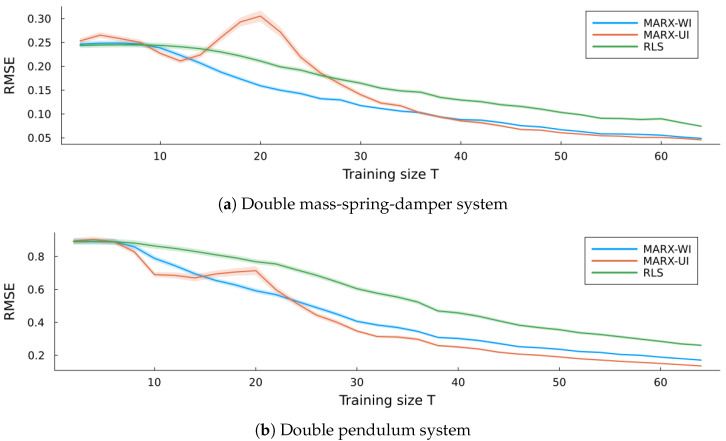
Simulation errors (average RMSE) of all three estimators for each validation system, with ribbons indicating standard errors.

**Figure 13 entropy-27-00679-f013:**
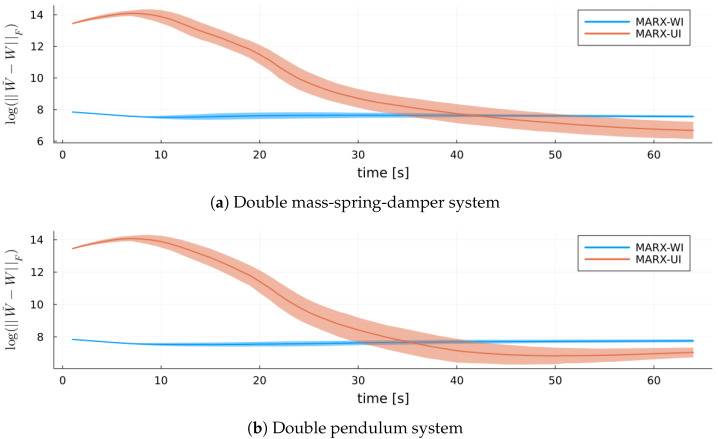
Log-scale Frobenius norm of the error between the true coefficient matrix $\tilde{W}$ and its estimates *W* from each MARX estimator for each validation system. Ribbons represent standard errors.

**Figure 14 entropy-27-00679-f014:**
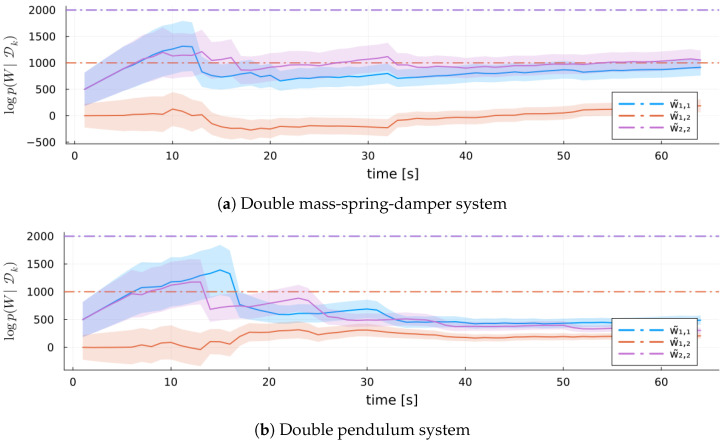
Time series of $\tilde{W}$ estimates from MARX-WI for each validation system, with ribbons representing one standard deviation. Horizontal lines mark true parameter values.

**Figure 15 entropy-27-00679-f015:**
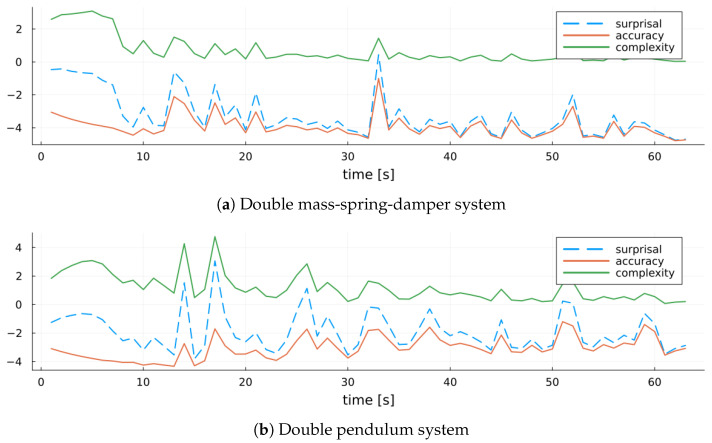
Surprisal (dashed blue) and its decomposition into accuracy (red) and complexity (green) for MARX-WI over time for each validation system.

**Figure 16 entropy-27-00679-f016:**
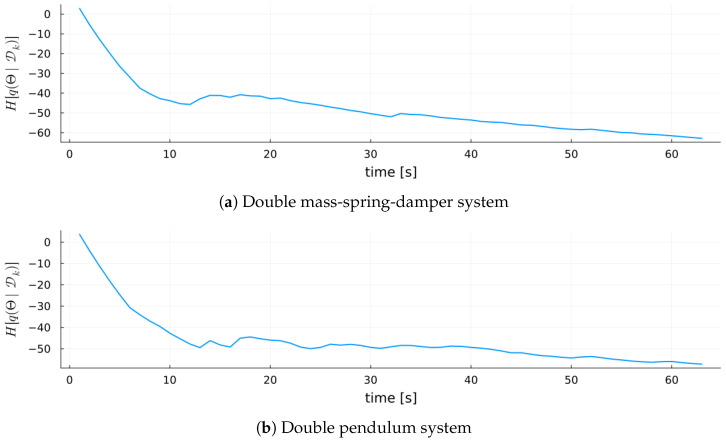
Entropy of the MARX-WI model parameters over time for each validation system.

**Table 1 entropy-27-00679-t001:** Sets of prior parameters used in the experiments.

	$M_{0}$	$\Lambda_{0}$	$\Omega_{0}$	$\nu_{0}$
Uninformative	$0_{D_{x}\times D_{y}}$	$1\times 10^{-4}\cdot I_{D_{x}}$	$1\times 10^{-5}\cdot I_{D_{y}}$	$D_{y}+3$
Weakly informative	$0_{D_{x}\times D_{y}}$	$1\times 10^{-1}\cdot I_{D_{x}}$	$1\times 10^{-2}\cdot I_{D_{y}}$	$D_{y}+3$

## Data Availability

All data in this work is synthetic. For details on it was simulated, see the accompanying repository at https://github.com/biaslab/MDPI2025-MARX (accessed on 8 March 2025).

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
