# Peer review of "Factor Graph-Based Online Bayesian Identification and Component Evaluation for Multivariate Autoregressive Exogenous Input Models [Author-notes fn1-entropy-27-00679]"

_entropy, 2025, doi:10.3390/e27070679_

Round 1
Reviewer 1 Report
Comments and Suggestions for Authors
"Graph Identification and Component Based on Online Graphs
evaluation for multivariate autoregressive exogenous input models" is composed of a very interesting, rigorous, and scientifically scrupulous text. We believe that the following modifications would improve reading conditions:
- As the number of acronyms is significant, it would be appropriate to add an appendix with a list of acronyms.
-At the end of the Introduction, a brief description of the structure of the rest of the text must be added. In other words, list the remaining sections, indicating what is done in each one.
-The "7. Discussion" section should be renamed "7. Addressing Performance". And your last paragraph should move into the next section.
-The "8. Conclusions" section is fine, with the exception of only mentioning positive situations. It is also necessary to take a critical look at the research and identify weaknesses that can provide clues for future research. An example is the text in section 7, which is recommended to be moved to section 8, duly worked on.
- Check the situations in which the word "whose" is used. in some cases you must use "which". "whose" is for people and "which" is for things.
See report.
Author Response
We thank Reviewer #1 for the careful review and insightful comments. Attached we provide a detailed, point-by-point response in the form of a PDF document. Please see the updated manuscript for changes (in blue).

Reviewer 2 Report
Comments and Suggestions for Authors
The paper should be thoroughly improved before it can be considered for publication. Despite providing an interesting approach to the concept of state-space models, the methodology and experiments should be made more clear. More importantly, the inference section is confusing and undermines the paper’s credibility.

Author Response
We thank Reviewer #2 for the careful review and insightful comments. Attached we provide a detailed, point-by-point response in the form of a PDF document. Please see the updated manuscript for changes (in blue).

Round 2
Reviewer 2 Report
Comments and Suggestions for Authors
The revised manuscript constitutes a substantial improvement and is progressing in the right direction. Most of my previous comments have been addressed satisfyingly. However, the primary concern remains that the selection of the application scenario, since it appears elementary. More applications and simulation results would definitely increase the methods's appeal.

Author Response
We thank the reviewer for the careful review and insightful comments. Attached you will find our rebuttal, please see the modified manuscript.

Round 3
Reviewer 2 Report
Comments and Suggestions for Authors
Again, the revised manuscript showed a substantial improvement. My previous comments have been addressed satisfyingly. An additional grammar revision before the paper's acceptance would be important.

Author Response
We thank Reviewer #1 for the careful review and insightful comments. Attached, we provide a detailed, point-by-point response. Please see the revised manuscript.
